# Using the Halophyte *Crithmum maritimum* in Green Roofs for Sustainable Urban Horticulture: Effect of Substrate and Nutrient Content Analysis including Potentially Toxic Elements

**Aikaterini N. Martini** [1,*]**, Maria Papafotiou** [1] **, Ioannis Massas** [2] **and Nikoleta Chorianopoulou** [1]

1   Laboratory of Floriculture and Landscape Architecture, Department of Crop Science, School of Plant Science, Agricultural University of Athens, Iera Odos 75, 118 55 Athens, Greece; mpapaf@aua.gr (M.P.); nickyxwr@gmail.com (N.C.)

2   Laboratory of Soil Science and Agricultural Chemistry, Department of Natural Resources and Agricultural Engineering, School of Plant Science, Agricultural University of Athens, Iera Odos 75, 118 55 Athens, Greece; massas@aua.gr

*   Correspondence: martini_agr@yahoo.com; Tel.: +30-210-5294-555

**Abstract:** The effect of substrate type and cultivation site in the urban fabric on growth, nutrient content and potentially toxic element (PTE) accumulation in tissues of the halophyte *Crithmum maritimum* was studied. Plantlets were cultivated for twelve months in containers with a green-roof infrastructure fitted and placed either on an urban second-floor roof or on ground level by the side of a moderate-traffic street. Two substrate types were used; one comprising grape marc compost, perlite and pumice (3:3:4, *v/v*) and one composed of grape marc compost, perlite, pumice and soil (3:3:2:2, *v/v*), with 10 cm depth. Plants grew well on both sites, although aboveground growth parameters and nutrient content in leaves were greater at street level. Both cultivation site and substrate type affected heavy-metal accumulation in plant tissues. Cu, Ni and Fe concentrations in leaves and Pb in roots were higher in street-level-grown plants compared to the roof-grown plants, and concentrations of Cu and Mn in leaves and Fe in both leaves and roots were lower in the soilless substrate compared to the soil-substrate, making the soilless type preferable in the interest of both safer produce for human consumption and lower construction weight in the case of green-roof cultivation.

**Keywords:** native ornamental plant; soil vs. soilless substrate; succulent plant; culinary and salad plant; urban agriculture; food safety; heavy metals in leaves and roots; nutrients in leaves

## 1. Introduction

The need for the planning of more sustainable cities leads to a wider use of extensive green roofs, as they can be applied to city buildings and provide significant advantages (environmental, social and economic) to the urban environment, mitigating the adverse effects of urbanization [1–4]. Considering the rapid increment in the built-up and low availability of open spaces for intensive green infrastructure in modern cities, green roofs are the most useful tool for urban greening, delivering multiple ecosystem services at the same time [5]. Apart from playing an important role in water retention, biodiversity increase and stormwater runoff management, urban heat-island effects reduce and building temperature regulation, and green roofs also constitute the missing link between the built and the natural environment [1,2,4,6]. Moreover, there is a growing interest in the utilization of urban rooftops as areas for food production, offering alternative spaces to grow vegetable products for urban markets, resulting in the facilitation of agricultural sustainability in urban areas [7].

Urban horticulture represents an opportunity for improving food supply, local economy, social integration and environmental sustainability within cities altogether, and leads to ecological benefits by reducing city waste, improving urban biodiversity and air quality, and overall, reducing the environmental impact related to both food transport and

storage [8,9]. Especially during the pandemic conditions of COVID-19, urban horticulture offers a more consistent food supply, prevents market disruptions and helps with stabilizing food prices, while it helps people become physically stronger and spiritually enriched. Recreational and aesthetic value to urban landscapes and individual homeowners is also offered [9]. Moreover, interactions between humans and plants in urban horticulture are considered to contribute to the good health and wellbeing of people, relieving physical and mental stress caused by urbanization through recently established "nature therapy" [10]. However, the high cost of urban land, along with high water and fertilizer needs, as well as risks to human health associated with food contamination by air and soil pollution can be inhibitory factors for urban horticulture [8,11]. Cultivation on building rooftops overcomes the issue of expensive or polluted land and provides a number of benefits for the planning of future sustainable cities [8].

By selecting the appropriate plant species for rooftop horticulture, the system could function as a green roof providing all the ecological and economic benefits that green roofs offer to the urban environment. Prior to the application of green roofs in arid or semi-arid areas, as in the eastern Mediterranean area, important factors to consider are water availability, biodiversity and local sustainability, conditions that can be met by the use of native plant species with limited water needs [3,12–16]. The adaptations of many Mediterranean plants to drought stress and their floristic diversity constitute them as ideal for extensive green roofs in cities with Mediterranean climates, thus also increasing urban biodiversity [3]. Moreover, halophytic plants constitute a viable solution in the landscaping of salt-affected soils, or when lower-quality water is used for irrigation [17]. This has been proved for *C. maritimum*, which was irrigated with recycled wastewater or reverse osmosis brine without any negative influence in its development [18]. Local vegetables and aromatic–medicinal species with low water demands could be ideal for combining green-roof technology with roof horticulture. However, it should be investigated whether it is safe for human health to consume such products, as there are risks associated with contamination by air pollutants in modern large cities.

The rapid technological and industrial development combined with the intensive exploitation of raw materials and energy sources has led to a steady global increase in the concentrations of toxic elements in the soil, water and air, which pose an increasing nutritional risk [19,20]. The presence of potentially toxic elements (PTEs) in the atmosphere, soil and water, even in traces, can cause serious problems in all organisms, while their bioaccumulation in the food chain can be particularly dangerous to human health [21]. Within the urban fabric, the higher overall traffic increases the trace metal content in crop biomass, while the presence of barriers between the growing area and the roads can significantly reduce the trace metal content [22]. Crop proximity to pollution sources increases heavy-metal accumulation in plant tissues, while soilless planting systems enable its reduction [23]. Moreover, the roadside rural environment can be significantly influenced by the proximity of a road because of road-dust particles. As a result, soil concentrations of Cr and Ni, as well as levels of all metals in Chinese cabbage near the road were found to increase [24]. Culinary herbs grown on artificial substrates on the land surface next to a moderate traffic road or a nearby green roof, although differing in the accumulation of heavy metals in their tissues, were found in both areas to contain higher concentrations of Ni and Pb than permissible levels for tissues of edible plants [25,26].

Heavy metals are persistent environmental pollutants deposited on surfaces and then adsorbed on vegetable tissues, either by deposits on various plant parts or by contaminated soil [27]. Atmospheric heavy metals may be absorbed via foliar organs of plants (cuticular cracks and stomatal pores) after wet or dry deposition of atmospheric fallouts on the plant canopy [22,28]. Heavy metals interfere with both the physiological activities of plants, such as photosynthesis, gaseous exchange and nutrient absorption, causing reductions in plant growth, dry matter accumulation and yield, and the levels of antioxidants in plants, reducing the nutritive value of the produce [29].

The extent to which heavy-metal contamination affects the safety of vegetable use depends on a number of factors, including the type of vegetable, the part of the plant being consumed and the type of heavy metal [30,31]. Usually, the vegetables of which the fruits and inflorescences are eaten accumulate lower amounts of heavy metals than the leafy or rhizomatous types [31,32]. The high variability in the trace metal content in various types of vegetables and places emphasizes the importance of specialized monitoring per crop and per place, to assess the possible effects on human health [22].

Eating the raw plant tissue of medicinal plants grown in polluted environments can also cause serious health problems [33]. In fact, it has been found that boiling the medicinal plant in water leads to the extraction of higher levels of the metal than immersion in hot water [34]. Various Egyptian spices and medicinal plants have been found to exceed the maximum permissible levels of heavy metals, depending on the plant species [34], while the corresponding plants grown under common field conditions in different regions of Austria contained the usual low ranges of heavy metals for plant material [35].

*Crithmum maritimum* L. (Apiaceae), known as sea fennel or rock samphire, is a widespread perennial succulent halophyte, up to 30–60 cm tall with fleshy leaves and umbels of pale yellow flowers (July-October), that usually grows on maritime rocks and sandy beaches along the Mediterranean and Black Sea coasts, as well as along the Atlantic coast of Portugal and south and south-west England, Wales and Southern Ireland [36–38]. Due to its characteristic taste and medicinal properties, it has been used since ancient times in the human diet [36] as an anticorrosive, diuretic, antioxidant and anti-inflammatory, as well as a spice. The stems, leaves and seed pods may be pickled in hot, salted, spiced vinegar, or the fresh leaves can be used in salads or to prepare soups and sauces or seasoning, especially for fish-based dishes [39]. Dried sea fennel, despite its reduced content of essential oils and chlorophylls, can be usefully exploited as a new food product both for its aromatic traits and as a natural colorant [40]. The introduction of dried sea fennel as a new spice-colorant in gastronomy could increase the sensory appeal of some traditional dishes and support the creation of many new recipes ([39]. Its roots, leaves and fruits are rich in several bioactive substances that exhibit antioxidant, antimicrobial [41–44] and insecticidal activities [45,46]. Moreover, this species is considered as an emerging crop, since it is not only a refined food and an interesting source of human health compounds and crop protection products, but also an alternative and sustainable crop suitable for saline agriculture [47–49].

*C. maritimum* is an ideal plant to combine the aesthetic and environmental principles of the green roof with rooftop horticulture in arid/semi-arid areas, because apart from its ornamental, horizontally spreading canopy, it exhibits heat tolerance and limited water demands [50]. Moreover, being a halophyte, it can overcome the salinity problems faced by rooftop horticulture [8]. The shallow substrate layer of an extensive green roof is susceptible to a build-up of salts that may be introduced by irrigation water [51]. *C. maritimum* growth has been evaluated in green-roof systems along with other Mediterranean xerophytes, mainly in artificial soilless substrates [12,15,50,52,53]. In Athens (Greece), it was successfully established on an extensive green roof either on a soilless- or on a soil-containing substrate, with a depth of only 7.5 cm [54,55].

The aim of this study was to evaluate *C. maritimum* growth and its safety for human consumption when cultivated at different sites and substrates in the urban fabric. The effects of green-roof versus roadside cultivation and soilless- versus soil-containing substrate on plant growth, along with PTE accumulation and nutrient content in foliage and root tissues, was studied over two harvesting periods. Washing of foliage was also examined as a way to remove surface deposition of environmental pollutants and thus affect the accumulation of PTEs and the product's safety for consumption.

## 2. Materials and Methods

### 2.1. Experimental Set-Up (Plant Material, Cultivation System and Site)

Four-month-old plantlets of *C. maritimum* produced by rhizome segments (approximately 5 cm long), were planted at the end of June 2016 in plastic containers 40 cm × 60 cm × 22 cm (two plants per container planted diagonally). The containers had a green-roof infrastructure fitted (moisture retention and protection of the insulation mat FLW-500, drainage layer Diadrain-25H and filter sheet VLF-150; Landco Ltd., green-roof systems Diadem, Athens, Greece). This cultivation system has been tested in previous studies on green roofs [13,16,25,26,56].

The containers were placed at two cultivation sites at the Agricultural University of Athens, half of them on the fully exposed second-floor flat roof of a building adjacent to the street Iera Odos (approximate height 7 m, approximate distance of the building from the street 12 m), and the other half at ground level in an open field next to the street Iera Odos (pavement width 1.5 m). Iera Odos is a moderate traffic road in the city of Athens, Greece. No buildings or vegetation served as barriers to traffic-related pollutants between the street and the cultivation sites.

A factorial experiment with two factors, i.e., cultivation site (green roof, street level) and substrate type (with soil, soilless) was undertaken. Thus, four treatments were applied (two cultivation sites × two substrate types), and in each treatment six containers were used, with two plants per container (*n* values are shown in data tables and figures). The containers were arranged following the completely randomized design.

### 2.2. Substrate

Two types of substrate were used; one comprising grape marc compost (C), perlite (Pe) and pumice (Pu) (3:3:4, *v/v*, soilless substrate) and one composed of grape marc compost, perlite, pumice and soil (S) (3:3:2:2, *v/v*, soil-substrate), with 10 cm depth. The grape marc compost was produced at the field of the Agricultural University of Athens ($37°59'$ N, $23°42'$ E) with the following procedure, which is routinely used for composting grape marc in Greece. Compost piles of trapezoid profile (2.5 m base width, 1.5 m top width, 1 m height, and 10 to 15 m length) were made in September to October. The piles were turned over every 2 to 3 weeks for the first 3 months. The humidity of the pile was maintained at over 50% by the natural rain events (rain events occur quite often during the fall–winter period in Greece). After this procedure, the compost was mature in spring [13]. The grape marc compost had a pH of 6.45 and EC 1155 μS/cm, the perlite particles were 1–5 mm in diameter (Perloflor; ISOCON S.A., Athens, Greece); the pumice particles were 1–8 mm in diameter (LAVA Mining and Quarrying Co., Paiania, Attiki, Greece) and the soil contained 21.4% clay, 25.8% silt, 52.8% sand and 21.32% equivalent $CaCO_3$, and had a pH of 7.9 and EC 241 μS/cm. The chemical composition of the substrates' components are presented in Table 1. The two substrates had similar pH values (7.5–7.6), while the EC value was 267 μS/cm for the soil-substrate and 352 μS/cm for the soilless type. Detailed physicochemical properties of the substrates are given in [56].

**Table 1.** Concentration (mg/kg) of heavy metals and content of nutrients in the substrate components.

|  | Cu * | Pb * | Ni * | Mn * | Zn * | Fe * | N | P-Olsen | $K_{exch}$ | $Na_{exch}$ |
|---|---|---|---|---|---|---|---|---|---|---|
| Soil | 0.159 | 0.459 | 0.038 | 1.222 | 0.19 | 0.74 | 0.091 | 11.52 ** | 60 ** | 380 ** |
| Grape marc compost | 0.248 | 0.759 | 0.045 | 0.428 | 0.213 | 4.646 | 2.814 | 0.5 | 1.96 | 0.16 |

Cu = copper; Pb = lead; Ni = nickel; Mn = manganese; Zn = zinc; Fe = iron; N = nitrogen; P = phosphorus; K = potassium; Na = sodium; * Heavy-metal concentrations extractable by DPTA; Total concentrations (%) of nutrients presented, excepting ** (mg/kg).

### 2.3. Irrigation

Irrigation was applied during the warm–dry period, i.e., April–October. Automatic drip irrigation on the substrate surface was applied before sunrise by two drippers placed

at equal distances from the center of the container and the plants (dripper supply 4 L·h$^{-1}$, irrigation period: 35 min), adequate to allow water to drain off the container

Plants were irrigated when substrate moisture was 17–20% *v/v*. In the first week of each month, substrate moisture (% *v/v*) was recorded daily to test the need for resetting the irrigation schedule. Three measurements from each container at 1900 to 2000 HR were taken using a handheld moisture meter (HH2; Delta-T devices, Cambridge, UK), with a soil moisture dielectric sensor (WET-2; Delta-T devices) inserted from the surface, which measured 65 mm in depth and 45 mm in width. Thus, irrigation was scheduled every 4 days in April–mid July and September–October and every 3 days mid-July–August.

### 2.4. Meteorological Data

The monthly ambient average, as well as maximum and minimum air temperature, total rainfall and average wind speed (http://meteosearch.meteo.gr/, accessed on 7 March 2022), the monthly average relative humidity and total radiation (Laboratory of General and Agricultural Meteorology, Agricultural University of Athens) and the monthly total sunshine duration (http://www.emy.gr/emy/el/climatology/climatology, accessed on 30 March 2022) during the experimental period (June 2016 to June 2017) are presented in Figure 1.

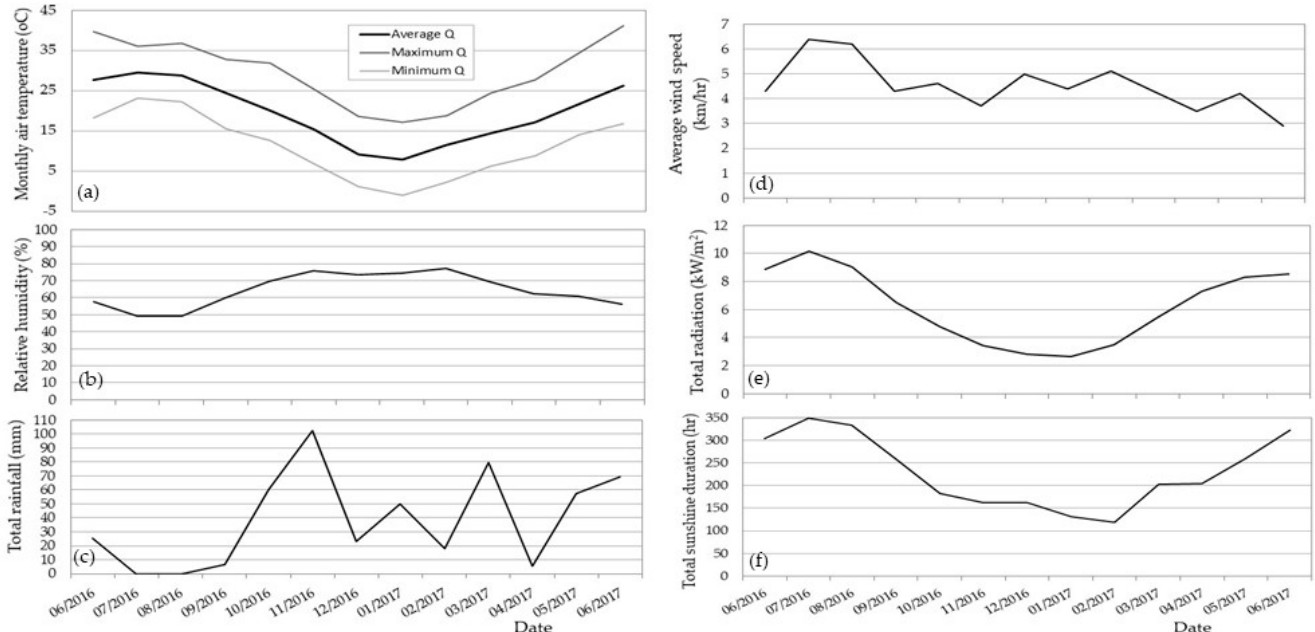

**Figure 1.** Average, maximum and minimum monthly air temperature (**a**), average monthly relative humidity (**b**), total monthly rainfall (**c**), average monthly wind speed (**d**), total monthly radiation (**e**) and total monthly sunshine duration (**f**), during the experimental period from June 2016 to June 2017.

### 2.5. Plant-Growth Evaluation

Plant growth was evaluated in September 2016 and June 2017, measuring canopy height and horizontal diameter (average value of greater horizontal diameter and its perpendicular), which led to the calculation of the plant canopy "growth index" (average of height and the two diameters). At the end of the experiment (June 2017), fresh and dry weights of the aboveground parts and the root systems were also recorded (see below in Section 2.6).

### 2.6. PTE and Nutrient Determination

In early September 2016, all shoots were pruned at about 5 cm height, and plants were left to sprout. The shoots collected were used in order to determine the heavy-metal accumulation and nutrient content in the leaves that constituted the edible part of the plant.

At the end of June 2017, the aboveground parts of the plants were collected in order to determine once more the heavy-metal accumulation and nutrient concentrations in the leaves, as in this period (early summer) they are collected for human consumption. The root systems of the plants were also removed and rinsed under running tap water in a colander to reduce root loss. Roots of both plants of each container constituted one sample, because roots were tangled and difficult to separate. Fresh weight of aboveground part and roots was measured immediately after their collection. Then, half of the canopy samples of each treatment were dipped in distilled water for 1 min and then rinsed under running tap water to wash off the dust deposited on the leaves This was carried out to check whether washing could reduce the possible concentrations of heavy metals, in case heavy metals were found on both the leaf surface and inside the leaf tissue. The samples were then dried in a dryer device at 60 °C for 7 days, and their dry weight was measured. In the dried canopy samples, leaves were removed from shoots and used for further analysis. Samples of washed and unwashed leaves and samples of roots were crushed and ground in a mill (Retsch ZM1000, Apeldoorn, The Netherlands), followed by sieving through a 0.5-mm sieve. Then, they were placed in individual airtight plastic bags and were kept in the refrigerator until the analysis.

For the determination of heavy metals, a certain quantity (1 g) of dried plant sample was placed on a porcelain crucible in a muffle furnace (at 550 °C for 3 h); in the combustion product, 5 mL $HNO_3$ (65%) was added, the solution was filtered and finally the filtrate was diluted with distilled water to a certain volume (25 mL). Concentrations of the heavy metals copper (Cu), lead (Pb), nickel (Ni), manganese (Mn), zinc (Zn) and iron (Fe) in samples were determined by atomic absorption spectrophotometry using a Varian-Spectra A300 system. In order to deduce whether it was safe to consume the leaves, the recorded heavy-metal concentrations were compared with the maximum acceptable limits for each heavy-metal concentration in edible plants, that is, 40.0 for Cu, 5.0 for Pb, 2.0 for Ni, 30.0 for Mn and 60.0 for Zn (mg/kg dry matter), according to FAO/WHO [57].Nitrogen (N) content of plant samples was determined by the Kjeldahl method in the Bucchi device [58]. Phosphorus content of plant samples was determined by a Shimadzu UV-1700 spectrophotometer. For every 10 samples a control sample was analyzed, and at the end of the measurement procedure, 30% of the samples were reanalyzed to test reproducibility. Exchangeable potassium (K) and sodium (Na) concentrations were quantified using a PGI 2000 flame photometer (PG Instruments Ltd., Leicestershire, UK).

### 2.7. Statistical Analysis

The data followed the normal distribution. The significance of the experiment was tested by one-, two- or three-way analysis of variance (ANOVA), and the treatment means were compared by Student's $t$ test at $p \leq 0.05$ (JMP 13.0 software, SAS Institute Inc., Cary, NC, USA, 2013).

### 3. Results

### 3.1. Plant Growth

Two months after their establishment, in early September 2016, plants grown at the level of the street developed larger canopies than those grown on the roof (Figure 2a). This result was also expressed in the fresh, but not the dry weights of the canopies. As for the substrate type, it did not affect the canopy size, although the soilless type induced larger fresh and dry weight (Figure 2b,c).

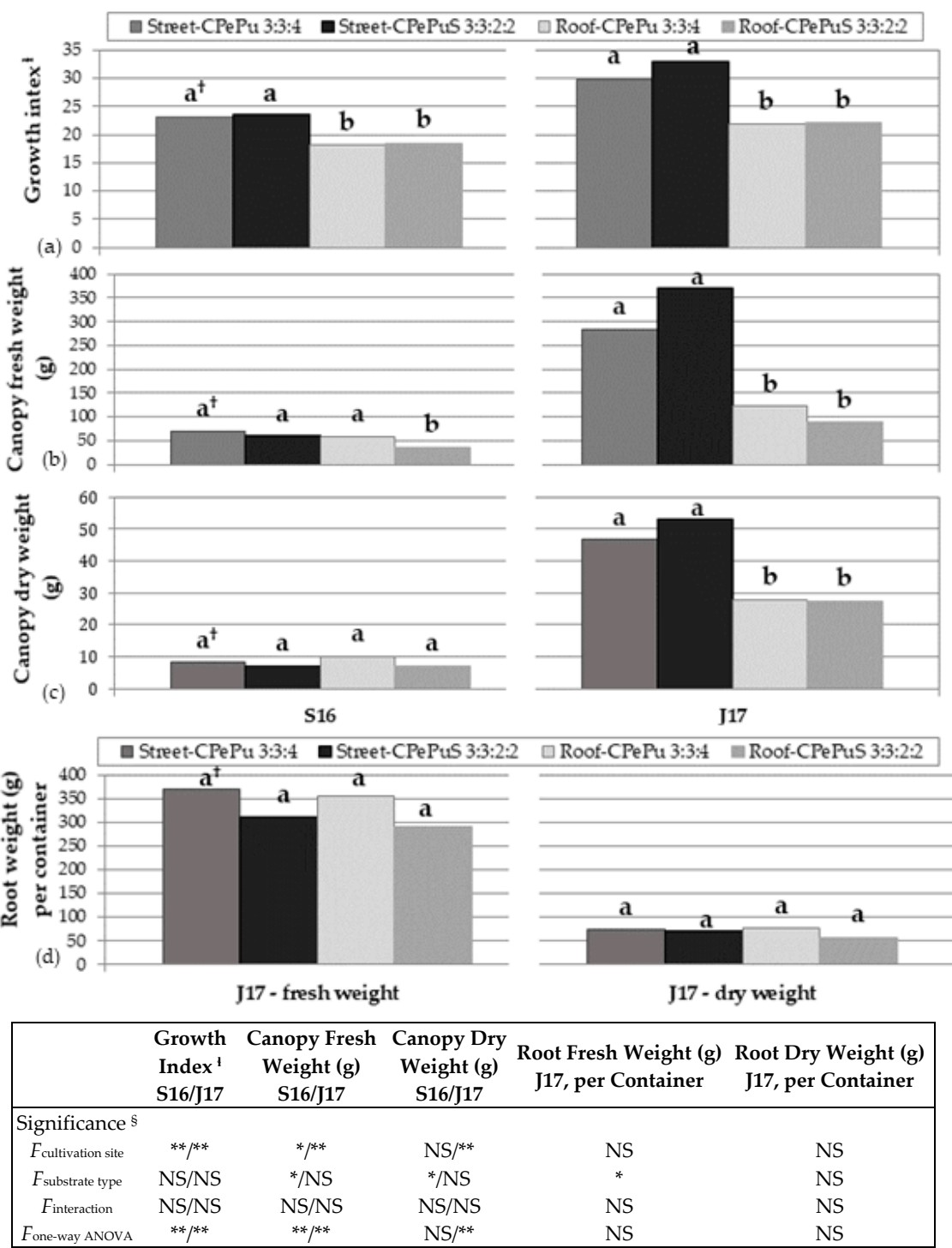

| | Growth Index [1] S16/J17 | Canopy Fresh Weight (g) S16/J17 | Canopy Dry Weight (g) S16/J17 | Root Fresh Weight (g) J17, per Container | Root Dry Weight (g) J17, per Container |
|---|---|---|---|---|---|
| Significance [§] | | | | | |
| $F_{cultivation\ site}$ | **/** | */** | NS/** | NS | NS |
| $F_{substrate\ type}$ | NS/NS | */NS | */NS | * | NS |
| $F_{interaction}$ | NS/NS | NS/NS | NS/NS | NS | NS |
| $F_{one-way\ ANOVA}$ | **/** | **/** | NS/** | NS | NS |

[†] Mean values (*n* = 12, excepting roots in which *n* = 6) in each column followed by the same lower-case letter do not differ significantly at $p \le 0.05$ by Student's *t* test; [§] NS or * or **, non-significant at $p \le 0.05$ or significant at $p \le 0.05$ or $p \le 0.01$, respectively; [1] Growth index = (plant height + diameter 1-greater horizontal diameter + diameter 2-perpendicular of diameter 1)/3; C: grape marc compost, Pe: perlite, Pu: pumice and S: soil.

**Figure 2.** Effect of cultivation site and substrate type on growth index (**a**), canopy fresh (**b**) and dry (**c**) weight, and root weight per container (**d**) of *C. maritimum*, after cultivation for two and twelve months in plastic containers with a green-roof infrastructure fitted (September 2016, S16 and June 2017, J17).

Soon after cutting the aboveground parts at the first harvest period, all plants sprouted showing stagnancy in their development until March 2017, probably due to the low temperatures of the period October–March (Figure 1a). Subsequently, the plants grew vigorously as the temperature rose, while at the same time there was high relative humidity and rainfall (Figure 1a–c). At the end of June 2017, twelve months after the establishment, plants cultured at the street level had greater canopy growth and higher fresh and dry weights of canopy compared to those cultured on the roof. Regarding the root systems, there were no differences in either the fresh or the dry weights of the roots (Figures 2 and 3).

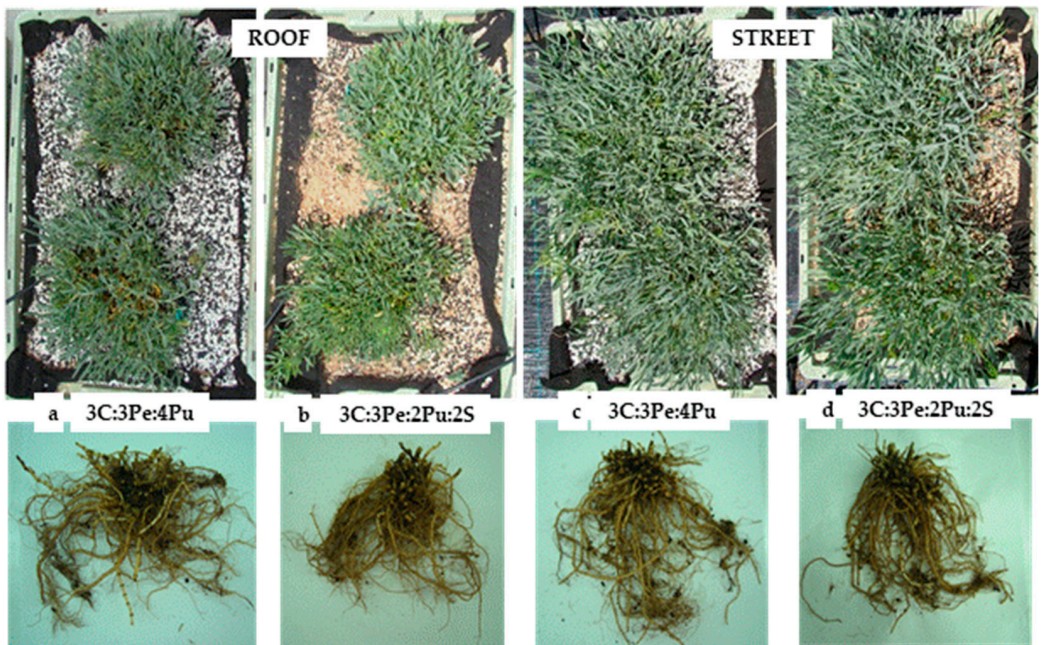

**Figure 3.** Characteristic canopy growth and root systems of *C. maritimum* in an extensive green-roof system, in the roof and the soilless (**a**) or soil (**b**) substrate, as well as in the street and the soilless (**c**) or soil (**d**) substrate, after 12 months of cultivation (June 2017).

*3.2. Heavy Metals*

Three-way ANOVA of heavy-metal concentrations in the leaves showed no interactions between substrate type, cultivation site and washing of the leaves (3-way ANOVA results not presented). Leaf washing had no significant effect on any heavy-metal concentration. Thus, concentrations of heavy metals were analyzed together for washed and unwashed leaves using two-way ANOVA, with the main factors being cultivation site and substrate type.

Chemical analysis of substrate components showed that all micronutrients (Cu, Mn, Fe, Zn) in soil, extracted by DTPA, were in deficiency, while concentrations of Pb and Ni were low and indicative of very low concentrations available for plant uptake (Table 1). After two months of cultivation, the site of cultivation affected only the accumulation of Cu and Zn in the leaves. Cultivation beside the street led to higher concentrations of Cu but lower accumulations of Zn compared to roof cultivation (Table 2). The substrate type affected only Pb and Mn accumulation, with the use of soil-substrate resulting in higher accumulations of these two heavy metals in the leaves (Table 2). Ni and Fe did not differ among experimental factors and treatments.

**Table 2.** Effect of cultivation site and substrate type on concentrations (mg/kg dry matter) of the heavy metals copper, lead, nickel, manganese, zinc and iron in leaves of *C. maritimum*, collected after cultivation for two (S16) and twelve (J17) months in plastic containers with a green-roof infrastructure fitted (data for washed and unwashed leaves are presented together).

| Cult. Site | Substrate Type (*v/v*) | Cu S16/J17 | Pb S16/J17 | Ni S16/J17 | Mn S16/J17 | Zn S16/J17 | Fe S16/J17 |
|---|---|---|---|---|---|---|---|
| Roof | 3C:3Pe:4Pu | 4.4 b [†]/4.5 b | 20.1 a$^z$/21.6 b$^z$ | 5.3 a$^z$/5.2 a$^z$ | 28.1 c/32.0 c$^z$ | 28.6 ab/35.2 a | 73.2 a/51.3 b |
| | 3C:3Pe:2Pu:2S | 5.2 ab/5.0 b | 23.0 a$^z$/24.5 a$^z$ | 5.9 a$^z$/5.8 a$^z$ | 35.6 b$^z$/38.7 b$^z$ | 33.6 a/32.6 a | 58.9 a/54.5 b |
| Street | 3C:3Pe:4Pu | 5.8 a/5.4 b | 20.6 a$^z$/24.0 a$^z$ | 5.8 a$^z$/5.9 a$^z$ | 25.8 c/32.9 bc$^z$ | 22.1 bc/31.2 a | 82.7 a/71.3 a |
| | 3C:3Pe:2Pu:2S | 6.4 a/6.5 a | 21.6 a$^z$/23.2 ab$^z$ | 5.6 a$^z$/6.0 a$^z$ | 41.9 a$^z$/45.9 a$^z$ | 20.6 c/38.1 a | 100.3 a/80.9 a |
| Significance [§] | | | | | | | |
| $F_{\text{cultivation site}}$ | | */** | NS/- | NS/* | NS/NS | **/NS | NS/** |
| $F_{\text{substrate type}}$ | | NS/* | */- | NS/NS | **/** | NS/ NS | NS/* |
| $F_{\text{interaction}}$ | | NS/NS | NS/** | NS/NS | NS/NS | NS/ NS | NS/NS |
| $F_{\text{one-way ANOVA}}$ | | */** | NS/* | NS/NS | **/** | **/ NS | NS/** |

[†] Mean values (*n* = 6) (±SE) in each column followed by the same lower-case letter do not differ significantly at $p \leq 0.05$ by Student's *t* test; [z] Mean concentration of the heavy metal higher than maximum acceptable limit for it according to FAO/WHO [57]; [§] NS or * or **, non-significant at $p \leq 0.05$ or significant at $p \leq 0.05$ or $p \leq 0.01$, respectively; C: grape marc compost, Pe: perlite, Pu: pumice and S: soil.

In all treatments, concentrations of Cu and Zn were lower than the maximum acceptable limits for these heavy-metals in edible plants according to FAO/WHO (2011), as opposed to concentrations of Pb and Ni, which were over the maximum acceptable limits. Regarding Mn, only samples from plants grown in the substrate with soil contained higher levels than the maximum acceptable limits (Table 2).

After twelve months of cultivation, concentrations of Cu, Ni and Fe were higher in leaves from plants grown at the street level than those on the roof, and Cu, Mn and Fe were higher in leaves from the soil-substrate than those from the soilless type (Table 2). With regard to treatments, Ni and Zn concentrations did not differ among treatments, whereas Cu concentration was highest in leaves from plants grown at the street level in the soil-substrate, and Pb concentration was lowest in leaves from plants grown on the roof in the soilless substrate (Table 2). Concentrations of Pb, Ni and Mn in all treatments were over the maximum acceptable limits for consumption by humans (Table 2).

In roots, cultivation at street level led to higher concentrations of Pb and Zn compared to the roof cultivation, and soilless substrate led to higher Ni and Zn than the soil-substrate, while the opposite was observed with Fe, which was higher in the soil-substrate than in the soilless type (Table 3). Cu, Ni and Mn were not affected by the treatments. The highest concentration of Pb was recorded in roots from plants grown at the street level in the soil-substrate, and the highest concentration of Zn occurred in roots from plants grown on the roof in the soilless substrate (Table 3).

Although *C. maritimum* roots are not consumed by humans, recorded concentrations of heavy metals were compared to their maximum acceptable limits, and those of Pb and Ni were found to exceed the limits (Table 3). Comparing roots and leaves, in roots, concentrations of Cu, Ni and particularly of Fe were higher, and concentrations of Mn and Zn were lower than in leaves, while there were no differences in Pb concentration (Tables 2 and 3).

**Table 3.** Effect of cultivation site and substrate type on concentrations (mg/kg dry matter) of the heavy metals copper, lead, nickel, manganese, zinc and iron, and content (%) of the nutrients nitrogen, phosphorus, potassium and sodium in the roots of *C. maritimum*, after cultivation for twelve months in plastic containers with a green-roof infrastructure fitted.

| Cultivation Site | Substrate Type (*v/v*) | Cu | Pb | Ni | Mn | Zn | Fe | N | P | K | Na |
|---|---|---|---|---|---|---|---|---|---|---|---|
| Roof | 3C:3Pe:4Pu | 9.3 a [†] | 20.6 bc[z] | 7.5 a[z] | 26.2 a | 30.8 a | 229.9 c | 0.8 a | 0.3 a | 2.5 a | 0.2 c |
| | 3C:3Pe:2Pu:2S | 7.5 a | 20.3 c[z] | 7.2 a[z] | 22.6 a | 24.4 b | 336.9 b | 0.7 a | 0.5 a | 2.9 a | 0.2 c |
| Street | 3C:3Pe:4Pu | 7.4 a | 22.3 b[z] | 10.0 a[z] | 24.7 a | 24.1 b | 233.0 c | 0.9 a | 0.3 a | 2.7 a | 0.4 a |
| | 3C:3Pe:2Pu:2S | 7.1 a | 24.4 a[z] | 6.6 a[z] | 26.3 a | 20.4 c | 454.2 a | 0.9 a | 0.3 a | 2.7 a | 0.3 b |
| Significance [§] | | | | | | | | | | | |
| $F_{cultivation\ site}$ | | NS | ** | NS | NS | ** | NS | * | NS | NS | ** |
| $F_{substrate\ type}$ | | NS | NS | * | NS | ** | ** | NS | NS | NS | NS |
| $F_{interaction}$ | | NS | NS | NS | NS | NS | NS | NS | NS | NS | NS |
| $F_{one-way\ ANOVA}$ | | NS | ** | NS | NS | ** | ** | NS | NS | NS | ** |

[†] Mean values (*n* = 3) (±SE) in each column followed by the same lower-case letter do not differ significantly at $p \leq 0.05$ by Student's *t* test; [z] mean concentration of the heavy metal higher than maximum acceptable limit for it according to FAO/WHO [57]; [§] NS or * or **, non-significant at $p \leq 0.05$ or significant at $p \leq 0.05$ or $p \leq 0.01$, respectively; C: grape marc compost, Pe: perlite, Pu: pumice and S: soil.

*3.3. Nutrient Content*

Regarding nutrient content in leaves and roots, the substrate type did not have any effect (Tables 3 and 4). After two months of cultivation, percentages of N, K and Na were higher in those from plants grown at the street level compared to leaves from plants grown on the roof, while the percentage of *p* was highest in leaves from plants grown on the roof in the soil-substrate (Table 4).

**Table 4.** Effect of cultivation site and substrate type on content (%) of the nutrients nitrogen, phosphorus, potassium and sodium in leaves of *C. maritimum* collected after cultivation for two (S16) and twelve (J17) months in plastic containers with a green-roof infrastructure fitted (data for washed and unwashed leaves are presented together).

| Cultivation Site | Substrate Type (*v/v*) | N S16/J17 | P S16/J17 | K S16/J17 | Na S16/J17 |
|---|---|---|---|---|---|
| Roof | 3C:3Pe:4Pu | 1.4 b [†]/1.0 b | 0.4 b/0.5 a | 4.6 b/5.4 a | 0.4 b/0.2 c |
| | 3C:3Pe:2Pu:2S | 1.5 b/1.2 ab | 0.5 a/0.5 a | 3.7 b/4.3 b | 0.4 b/0.2 c |
| Street | 3C:3Pe:4Pu | 2.1 a/1.6 a | 0.4 b/0.3 b | 7.7 a/5.1 a | 0.7 a/1.1 a |
| | 3C:3Pe:2Pu:2S | 2.2 a/1.5 a | 0.4 b/0.3 b | 7.2 a/5.3 a | 0.6 a/0.8 b |
| Significance [§] | | | | | |
| $F_{cultivation\ site}$ | | **/** | -/** | **/- | **/- |
| $F_{substrate\ type}$ | | NS/NS | -/NS | NS/- | NS/- |
| $F_{interaction}$ | | NS/NS | */NS | NS/** | NS/** |
| $F_{one-way\ ANOVA}$ | | **/* | **/** | **/** | **/** |

[†] Mean values (*n* = 6) (±SE) in each column followed by the same lower-case letter do not differ significantly at $p \leq 0.05$ by Student's *t* test; [§] NS or * or **, non-significant at $p \leq 0.05$ or significant at $p \leq 0.05$ or $p \leq 0.01$, respectively; C: grape marc compost, Pe: perlite, Pu: pumice and S: soil.

After twelve months of cultivation, plants grown at street level had higher N and lower *p* percentages in their leaves compared to those grown on the roof. N percentage was lowest in leaves from plants grown on the roof in the soilless substrate, while K percentage was lowest in leaves from plants grown on the roof in the soil-substrate (Table 4).

In roots, percentages of N and Na were higher in plants grown at the street level compared to those grown on the roof, while there were no differences in the percentages of N, P and K among treatments (Table 3). The content of nutrients in roots, excepting the content of P, were lower than the corresponding content in leaves (Tables 3 and 4).

## 4. Discussion

### 4.1. Plant Growth

Cultivation site affected plant growth, since plants placed at street level had greater all-growth parameters in the aboveground parts compared to those placed on the roof. The growth environment of green roofs is considered severe as a result of high exposure to solar radiation and wind and wide temperature fluctuations [2,3,12]. Therefore, the adverse conditions of the roof were probably responsible for the reduced plant growth, since aboveground plant biomass may be reduced under abiotic stress conditions [59–61]. Nevertheless, the difference in growth was not great, and plants grew satisfactorily on the roof (Figures 2 and 3). Although all plants sprouted in October 2016, after the first harvest period, they did not proceed in their development until March 2017, probably due to the low temperatures prevailing at this period.

Substrate type, although affecting plant growth at the establishment stage, verifying previous research [54], had no effect after one year of cultivation. This is in agreement with what has been reported for long-term cultivation of other Mediterranean xerophytes, such as *Origanum dictamnus*, *Pallenis maritima*, *Scabiosa cretica* and *Sideritis athoa*, cultivated on a green roof with the same types and depth of substrate as the present study [16,62].

The compact and horizontally spreading canopy of *C. maritimum* is a valuable feature for its uses in extensive green roofs, as the rapid ground cover contributes to better utilization of water provided during installation period, and subsequently to better water retention and stormwater management [63]. Being a succulent halophyte, it is suitable for use in extensive green roofs in arid/semiarid regions [50], possessing both drought and salinity tolerance; the shallow substrate of extensive green roofs is susceptible to a build-up of salts introduced by irrigation water [51].

Growth of root systems was not affected by either cultivation site or substrate type. Thus, while aboveground parts were reduced in plants grown in the green roof compared to those at the street level, their root systems did not differ. This resulted in a higher root-to-aboveground ratio in plants grown in the green roof, which is usual in a number of plant species, as under drought conditions a higher root/ aboveground ratio [64] optimizes water uptake [65].

Taking into account the long-term equal growth of the plants in both substrates, the soilless type could be suggested for green-roof cultivation in order to achieve low construction weight. This is in accordance with other studies on the species, in which lightweight and highly porous soilless substrates were recommended [15,50,52].

### 4.2. Heavy-Metal Concentration

The high concentrations of heavy metals found in leaf tissues of *C. maritimum*, especially of Pb, Ni and Mn, which exceeded permissible limits at both cultivation sites, verified previous research works at the same sites on *Salvia officinalis* and *Origanum vulgare* ssp. *hirtum*. In these plants also, concentrations of Pb and Ni in the leaves were higher than the permitted levels, either next to the road or on the green roof, independently of substrate type and fertilization applied [25,26].

Washing the leaves proved ineffective in the reduction of heavy-metal concentrations, being unable to reduce them to within the permissible limits. This shows that measured heavy-metal concentrations were not due to the dust that was kept by the foliage, but that deposited environmental pollutants had been absorbed by plant tissues. Bibliographic reports related to the effect of washing on concentrations of heavy metals mainly in leafy vegetables are contradictory, since some showed no difference between unwashed and washed leaves in the urban area [66,67], while others revealed higher concentrations of toxic elements in unwashed samples than in washed samples [21,24,68], indicating the deposition of traffic-related particles on the plant surface.

In roots, the cultivation site affected only the accumulation of Pb and Zn, Pb concentration being higher by the street, while Zn was higher on the roof. Substrate type affected Ni, Zn and Fe accumulation, Ni and Zn being higher in the soilless substrate while Fe was

higher in the soil-containing type. Fe concentration in roots was significantly higher than that recorded in leaves.

It is very difficult to distinguish whether metal concentrations within plant tissues are taken up by root cells from the substrate or by leaf surfaces from the atmosphere, because the two uptake pathways can occur simultaneously near urban and industrial areas, while they are affected by different factors [28]. Root metal uptake by plants depends on substrate and plant type, chemical speciation of metals in substrates, substrate particle size, cation-exchange capacity, pH, organic matter content and microbial activity, while foliar metal uptake is affected by several physical (type and chemistry of heavy metals, characteristics of plant leaf surface), chemical (metal speciation and cuticle composition) and biological (growth stage at which heavy metals are deposited on plant surface) factors [28,69]. Moreover, in the cases of root metal uptake and foliar metal uptake, the major portion of absorbed metals (more than 95%) is stored in the plant tissue that performed the uptake [28].

The increased concentrations of heavy metals found in plant tissues in the present work could be attributed to both the polluted urban atmosphere of Athens and the particular road traffic, including trucks, as there are a number of warehouses in the district. There were no recent reports found in the literature on Athens's atmosphere pollution. In the last decades (the period 1990–2009), a strong decreasing trend of pollutants was shown [70], while the concentrations of the elements Pb, Cd, Ni and Mn were not alarming [71]. Generally, road transportation is considered the major pollution source for the Greater Athens Area, while the amounts of pollutants emitted are determined by vehicles' ages and their corresponding engine technology, with vehicles aged more than 15 years ten years ago having been major polluters [70]. The financial crisis of 2010 in Greece increased the average age of cars in the country even more.

Overall, higher traffic density has been reported to increase trace metal content in crop biomass within the urban fabric, especially when the cultivation site was adjacent to a street without buildings or vegetation to serve as barriers to traffic-related pollutants [21–24,72]. In the present study, concentrations of PTEs in soil and gape marc compost used in the substrates were at very low concentrations available for plant uptake. Therefore, most plant-tissue contamination was due to the pollution of the atmosphere. Cultivation on the roof of a two-storey building located 12 m away from the street was only partly effective in significantly reducing levels of PTEs in plant tissues. Relative humidity has been shown to influence the permeability potential of the leaf surface and affect the physicochemical responses of plants to adsorbed PTEs concerning solubility or redox [28]. Although *C. maritimum* plant growth was restricted on the green roof compared to the ground-level cultivation, indicating that dry thermal conditions prevailed, the roof accumulation of Pb, Mn and Zn was not affected and that of Cu and Ni was only slightly reduced in the leaves, with only Fe accumulation presenting a significant reduction. Nevertheless, there are some reports showing that crop plants cultivated in urban gardens, including rooftop gardens, contain heavy metals within the permissible limits [21,66,68]. Moreover, aromatic crops can be grown as alternative high-value crops in metal-polluted areas and provide a metal-free marketable final product in the form of essential oil [73,74].

The use of vehicles of new technology, and especially of electric vehicles, constitutes a promising solution for reducing air pollution in cities [75–77]. However, a significant reduction in pollution (over 10%) can only be achieved when the total replacement rate reaches 50%, while both light and heavy vehicles will have to be replaced by electric ones [78]. If the use of electric vehicles expands in the near future, resulting in a reduction in atmosphere pollution in cities, airborne contamination of soils and crops is also expected to be reduced, thus making urban agriculture safer for humans.

### 4.3. Nutrient Content

Regarding the nutrient content of leaves, the effect of cultivation site was significant, as higher concentrations of N and Na were observed in cultivation at the street level, while concentrations of P were favored by the cultivation on the green roof. As for K, after two

months of cultivation, higher concentrations were measured at the street level, while after twelve months of cultivation, its concentration was found to be reduced on the green roof only in the soil-substrate. In the roots, only Na concentration was affected by the cultivation site and was favored by the cultivation at the street level. Comparing nutrient contents with a previous work on mineral-nutrient composition of wild *C. maritimum* leaves [79], N was found to be somewhat lower in our work and Na was significantly lower, while P and K were almost double in our study. *C. maritimum,* in its natural environment by the sea, is expected to have higher Na content, while increased K is evident in growth in adverse conditions, as it alleviates the harmful effects of abiotic stresses in plants [80].

## 5. Conclusions

*C. maritimum* grew well in the urban environment, both at the ground level next to a moderate traffic street and in an extensive green roof, proving that it is suitable for use in the urban fabric. Its cultivation at the ground level led to greater canopy growth compared to that in the green roof. Leaves from plants cultivated at the street level showed higher nutrient content, while in roots there were no significant differences among treatments.

Plants grew equally satisfactorily in both the soilless lightweight substrate and the soil-containing type, but the soilless substrate is the preferable option, firstly for lowering construction weight in the case of green-roof cultivation, and secondly for safer production for human consumption, as the soilless substrate resulted in lower Cu, Fe and Mn accumulation in the leaves.

Cultivation site also affected the accumulation of heavy metals in plant tissues, as Cu, Ni and Fe concentrations in the leaves and Pb in the roots were lower in green-roof grown plants than in those grown at the street level.

Values of Fe in roots were significantly greater than those recorded in leaves.

Washing of the leaves was ineffective in reducing concentrations of heavy metals.

*C. maritimum* is recommended for use in sustainable green roofs and urban horticulture, as it showed satisfactory growth and resistance to harsh green-roof conditions in lightweight and low-depth substrate. If human consumption of its edible parts is desirable, the site or location of cultivation should be carefully chosen to minimize contamination with environmental pollutants, ensuring food safety.

**Author Contributions:** Conceptualization, A.N.M. and M.P.; methodology, A.N.M., M.P. and I.M.; validation, A.N.M., M.P., I.M. and N.C.; formal analysis, A.N.M. and N.C.; investigation, A.N.M., M.P., I.M. and N.C.; resources, M.P. and I.M.; data curation, A.N.M. and N.C.; writing—original draft preparation, A.N.M., M.P. and I.M.; writing—review and editing, A.N.M., M.P. and I.M.; visualization, A.N.M.; supervision, M.P. All authors have read and agreed to the published version of the manuscript.

**Funding:** This research received no external funding.

**Informed Consent Statement:** Not applicable.

**Conflicts of Interest:** The authors declare no conflict of interest. The funders had no role in the design of the study; in the collection, analyses, or interpretation of data; in the writing of the manuscript, or in the decision to publish the results.

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
