# Peer review of "Using the Halophyte Crithmum maritimum in Green Roofs for Sustainable Urban Horticulture: Effect of Substrate and Nutrient Content Analysis including Potentially Toxic Elements"

_sustainability, doi:10.3390/su14084713_

Round 1
Reviewer 1 Report
In this paper, the effects of different locations and different substrate conditions on the metal content in salt-tolerant plants were compared through experiments. In general, the topic selection is more meaningful, the experimental method is suitable, and the logical structure is relatively clear, but some experimental conditions need to be further elaborated.
1, The detailed steps of grape pomace composting need to be explained;
2, The heavy metal content of the soil matrix needs to be examined to assess its effect on the metal content of Crithmum maritimum.
3, Some natural conditions of streets and roofs, such as light, wind, humidity, etc., also need to be measured and compared, and the influence of natural conditions on the experimental results should be analyzed.
Reviewer 2 Report
Knowledge generated is strictly confined to the testing of specific soil and soilless based substrates for nutrient content and toxic elemenrts, indicators for plant growing on ground level and second floor roof in an urban setting. The agriculture-oriented research suffers from the handicap of a comprehensive and integrated study involving the host - (1) architectural design, (2) roof construction details, (3) water retention and irrigation design, (4) the physical attributes to the microclimatic from an environmental perspectives, and (5) urban planning of traffic-prone kerbside or frontage, etc. to mitigate or control of the hazards directly affecting the substrate for plant growth. In this way, the finding presented so far could not do a innovative and better job by advising city managers, planners and building designers in this otherwise an collective endeavor.
World trends have shown a major shift of vehicules from petroleum to electric based. For majority of nations, it is evident to expect a transformational change - thus an anticipated disappearance of the environmental threats under studied in a matter of years rather than decades. The authors are invited to address this in their discussion part in order to advance their understanding of the other actors that have an important impact on urban agriculture.
Reviewer 3 Report
Overall, the article looks well structured and covers a topic of interest to readers. I only recommend modifying the title so that it describes exactly what the document is about, justifying the comparison of means in the statistical part and a few small errors in the writing (indicated in the PDF attached).

Round 2
Reviewer 2 Report
The revision has marginally improved because it does not managed to address the symbiotic relationship between the plant species & the soil as an integral part of the building and building system not to mention the role of architecture, and building construction involved. In other words, it does not recognized the importance or benefits it would otherwise achieve by addressing those technical issues imposed by roof as a provider for edible plants. Given its current stance, it is not convinced that the knowledge generated by the present study is helpful in any ways for advancing the research on architectural design, engineering and construction in the given context.
